# FORM FOLLOWS FUNCTION: TEXT-TO-TEXT CONDITIONAL GRAPH GENERATION BASED ON FUNCTIONAL REQUIREMENTS

## ABSTRACT

This work focuses on the novel problem setting of generating graphs conditioned on a description of the graph's functional requirements in a downstream task. We pose the problem as a text-to-text generation problem and focus on the approach of fine-tuning a pretrained large language model (LLM) to generate graphs. We propose an inductive bias which incorporates information about the structure of the graph into the LLM's generation process by incorporating message passing layers into an LLM's architecture. To evaluate our proposed method, we design a novel set of experiments using publicly available and widely studied molecule and knowledge graph data sets. Results suggest our proposed approach generates graphs which more closely meet the requested functional requirements, outperforming baselines developed on similar tasks by a statistically significant margin.

## 1 INTRODUCTION

Many concepts can be described by graphs; including molecules (20), abstract syntax trees (19), knowledge graphs (27), and project schedules (18). Each of these concepts is used in downstream tasks where graph structure has a direct impact on task performance. For example, some molecules can be used as medicine for a disease while others cannot and this is partially determined by the molecules' graphs. It would be useful if we could describe the functional requirements of a graph using natural language and query a model to conditionally generate a graph which meets these requirements. For example, a model using the prompt "generate a molecule with 47 valency electrons" would generate a valid molecule graph with 47 valency electrons. With such models we could speed up and improve the processes of drug discovery, software development, project management as well as many other applications of graphs.

In this work, we focus on the problem of generating graphs where node and edge features are strings of text and term this type of graph a *text graph*. Text graphs are a fairly flexible data format and can be used in most applications of graph-structured data including those listed above. To the best of our knowledge, text graph generation conditioned on text has only been studied in the fields of knowledge graph generation (8; 12; 14; 26; 27; 38) and explanation graph generation (33). In both setups the conditional text explicitly describes the graph. These explicit descriptions give an imperative "recipe" with step-by-step instructions of how to generate the graph. Contrary to giving explicit imperative descriptions, this work focuses on the case where the conditional text is a *functional* description of the graph, specifically for a downstream task. Additionally, in the case of imperative conditional text, there is an injective mapping between conditional text and the graph that should be generated, which means that methods can be evaluated by comparing the generated graph to the ground truth graph in terms of its structure. By contrast, many graphs might correspond to the same functional description and hence a new experimental design is required to study the problem.

We propose solving this problem by fine tuning a large language model (LLM) to generate a serialized text description of a graph. This is motivated by the strong performance pre-trained LLMs have shown when fine-tuned to perform a specific task (6; 17; 34), including the task of text-to-serialized graph generation (8). However, prior work on serialized graph generation does not generalize to all domains containing text graph structured data. To further this approach, we propose a serialization method which is expressive enough to reversibly serialize any graph and a method to incorporate

graph structure into the LLM's generation process motivated by the observation that the structures of generated graphs have a direct impact on their functional properties and consequently on model performance. While LLMs are probably expressive enough to learn to incorporate graph structure into their generation process, it is more efficient to provide the model with the ability to do so.

However, it is also a non-trivial challenge because modern LLMs perform autoregressive sampling to generate text (4; 5; 6; 17; 28; 31; 34). This involves a multi-step sequential process, where at each step a forward pass is performed and then a new token is sampled. A token is an atomic unit of text like 'dog', ' ', or 'ch'. The generation process exhibits high computational complexity and cost due to the requirement of performing a forward model pass for each generated token. Consequently, LLMs are typically trained to generate text without actually performing generation at training time. Only a single forward pass is performed at each training step without sampling and the entire sequence (conditional plus desired text) is fed into the LLM for that forward pass. As such, an LLM has access to the full sequence at training time, while at generation time it only has access to the conditional text and the part of the sequence that has already been generated. This means that during training an LLM could learn to rely on information that it will not have access to at generation time. This issue is overcome by using causal masking, which guarantees the LLM cannot pass information backwards in the token sequence ensuring that generation conditions are simulated at training time.

In Section 3, we propose a method of incorporating graph structure into an LLM's generation process which passes information between tokens in a sequence. Our key contribution is demonstrating how to do so without passing information backwards in the sequence. **Specifically, we propose to extend LLMs to process and generate sequences of text and graphs. For this we provide the LLM with the ability to deserialize graphs incorporated into input token sequences and introduce message passing layers into the LLM to calculate representations of graph structure in a manner that is conducive to autoregressive generation**.

In this work, we specifically focus on the problem of generating text graphs from functional requirements within the range of those seen at training time (interpolation), instead of extrapolating outside the training set's domain, which we leave as future work. We propose a novel experiment design to evaluate methods in this new problem setting using the publicly available data sets WebNLG+ 2020 (10) and PCQM4M (20). Results suggest that our proposed model outperforms previous work on text graph generation conditioned on text by a statistically significant margin. And that our proposed approach for incorporating graph structure into the language model's generation process leads to models which generate examples which on average more closely meet their conditional functional requirements. All code used to perform experiments is publicly available at ...

## 2 PRELIMINARIES

A text graph $\mathcal{G} = \{\mathcal{V}, \mathcal{E}\}$ is composed of a set of $N$ nodes $v_i \in \mathcal{V}$ each with an associated text string describing the node and a set of $M$ directed edges $(v_i, v_j) \in \mathcal{E}$ each with an associated text string describing the edge. Each text graph $\mathcal{G}$ is associated with text $\mathcal{D}_f$ containing a functional description of $\mathcal{G}$. Our goal is to train a model to accurately predict $p_{\boldsymbol{\theta}}(\mathcal{G}|\mathcal{D}_f)$ with parameters $\boldsymbol{\theta}$, describing the distribution over text graphs $\mathcal{G}$ conditioned on a specific functional description $\mathcal{D}_f$. Importantly, this distribution may be multimodal as multiple graphs may have the same functional properties.

This work focuses on the case where the parameterized model $p_{\boldsymbol{\theta}}(\cdot)$ is a language model that generates a serialized text description of the graph. Hence, if $\mathcal{D}_{\mathcal{G}}$ is the serialized graph, then the language model predicts $p_{\boldsymbol{\theta}}(\mathcal{D}_{\mathcal{G}}|\mathcal{D}_f)$. This requires an injective serialization function $g : \mathcal{G} \to \mathcal{D}_{\mathcal{G}}$ with a known inverse mapping $g^{-1} : \mathcal{D}_{\mathcal{G}} \to \mathcal{G}$ from the serialized description $\mathcal{D}_{\mathcal{G}}$ back to the graph $\mathcal{G}$ such that $\mathcal{G} = g^{-1}(g(\mathcal{G}))$. $g(\cdot)$ is used at training time to transform graphs $\mathcal{G}$ into serialized graphs $\mathcal{D}_{\mathcal{G}}$ and the deserialization function is used at generation time to recover the graph $\mathcal{G}$ from the serialized graph $\mathcal{D}_{\mathcal{G}}$ (see Figure 1 top row). With this approach, we pose the problem of generating graphs conditioned on functional descriptions as a text conditioned on text generation problem and can consequently use LLMs to conditionally generate text graphs based on functional descriptions.

Prior state-of-the-art work on serialized graph generation uses a `bag-of-edges` approach (8; 33), which describes each edge using a string with special tokens for delimiting graph elements and then lists out the edges in a predefined order. For example, the popular `smiles` molecule serialisation algorithm uses depth-first search traversal (39) to order chemical bonds (edges) in a

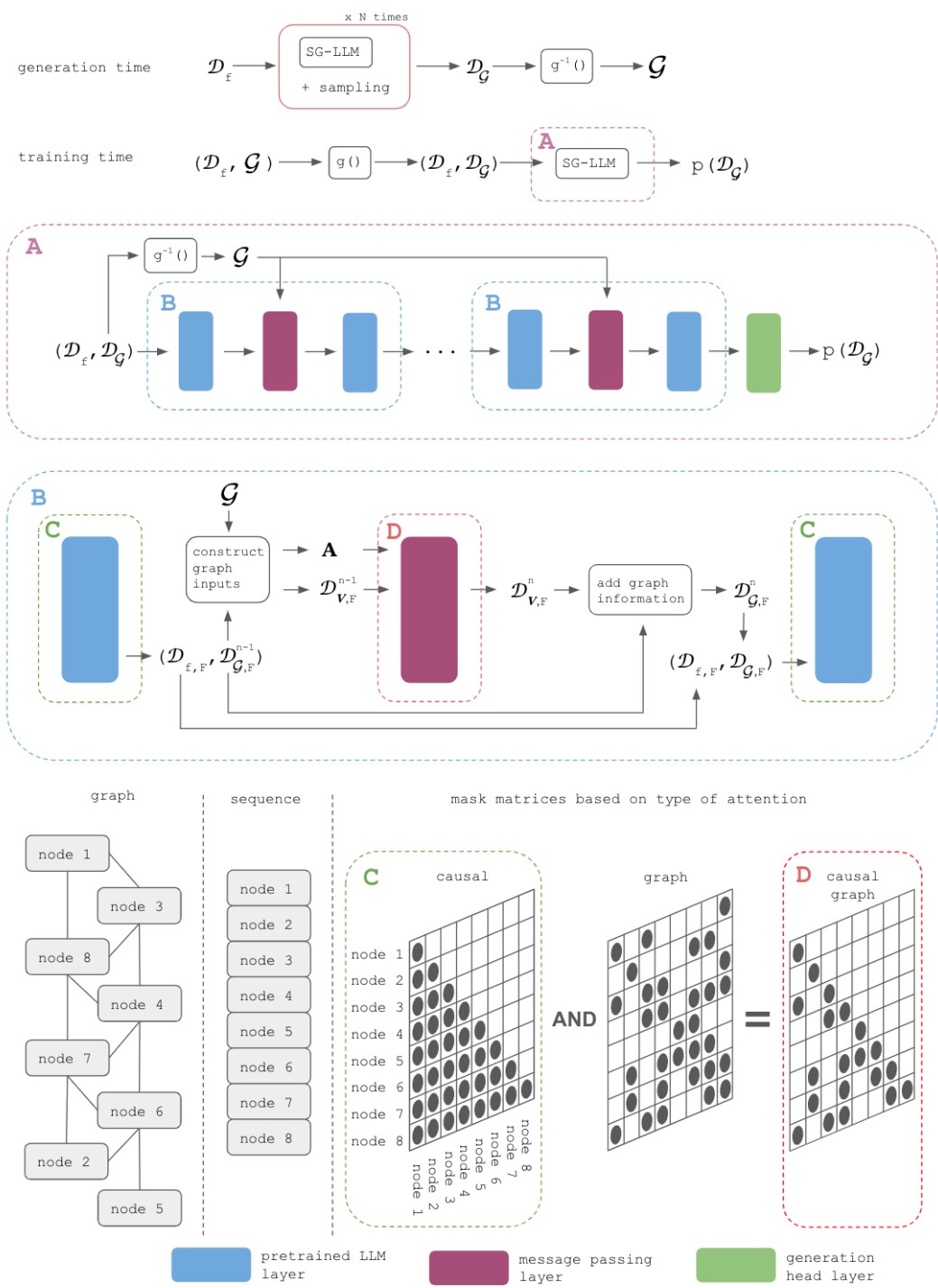

Figure 1: At generation time (top) the input to the model is the graph's functional description $\mathcal{D}_f$ and the output is a serialised description of the graph $\mathcal{D}_\mathcal{G}$. During training time (below top) the input is a functional description, serialized graph pair $(\mathcal{D}_f, \mathcal{D}_g)$; the output is the probability $p(\mathcal{D}_g)$. Graph serialization method $g(\cdot)$ is used to serialize input graphs before training, while the deserialization method $g^{-1}(\cdot)$ is used to deserialize the generated serialized graph $\mathcal{D}_g$ as well as to help perform message passing within the LLM. Block A describes the proposed architecture interleaving message passing layers between LLM layers. Block B depicts how information is passed between LLM and MP layers. The bottom visualization depicts the masked matrices used in the types of attention performed within LLM and message passing (MP) layers.

molecule. Then each edge is described with the syntax `<PN>predecessor node feature string<E>edge feature string<SN>successor node feature string`, where `<PN>,<E>,<SN>` are special tokens used to delimit the different components of the edge.

Using LLMs, a description $\mathcal{D}$ is represented in three ways: 1) as a string (text), 2) as a sequence of token indicators, and 3) as a sequence of feature vectors / embeddings. To distinguish between these three representations, we use $\mathcal{D}_S$ to denote the string representation, $\mathcal{D}_T$ to denote the sequence of token indicators and $\mathcal{D}_F$ to denote the sequence of feature vectors. For the serialized graph $\mathcal{D}_\mathcal{G}$, its string representation is denoted by $\mathcal{D}_{\mathcal{G},S}$, its sequence of token indicators representation by $\mathcal{D}_{\mathcal{G},T}$ and its sequence of feature vector representation by $\mathcal{D}_{\mathcal{G},F}$.

## 3 METHOD

The datasets LLMs are pre-trained on do not contain much, if any, data on generating serialized graphs (4; 6; 31; 34). Consequently, it is unlikely they could generate text graphs without further training. We provide evidence supporting this hypothesis in experiments where we evaluate a pretrained LLM without further fine-tuning. As shown in the results in Appendix D, an LLM without fine-tuning did not generate a single valid serialized graph. Hence, we propose fine-tuning an LLM on a training and validation set of (functional description, graph) pairs to generate serialized graphs conditioned on functional descriptions. While LLMs are probably expressive enough to learn to incorporate graph structure into their generation process, it is more efficient to provide the model with the ability to do so. State-of-the-art models for performing inference on graph structured data use message passing (MP) layers (15; 35). As such, we propose interleaving MP layers between language modelling layers in an LLM (Figure 1 block A). Any message passing layer can be used with our proposed approach [1].

### 3.1 FINE-TUNING OBJECTIVE

For fine-tuning, we propose minimizing the negative log-likelihood of serialized graph sequences $\mathcal{D}_\mathcal{G}$ as described in equation (1):

$$Loss = \frac{1}{\textcolor{red}{n_{batch}}} \sum_{i=1}^{n_{batch}} \left( -\log p_{\boldsymbol{\theta}}(d_{\mathcal{G},T}^{1,i}|\mathcal{D}_f^i) + \sum_{j=2}^{n_{seq}^i} -\log p_{\boldsymbol{\theta}}(d_{\mathcal{G},T}^{j,i}|d_{\mathcal{G},T}^{1:(j-1),i}, \mathcal{D}_f^i) \right) \quad (1)$$

where $d_{\mathcal{G},T}^{i,j} \in \mathcal{D}_{\mathcal{G},T}^i$, the superscript $i$ is the index in a batch, the superscript $j$ indicates the position of an element $d_{\mathcal{G},T}^{i,j}$ in the sequence $\mathcal{D}_{\mathcal{G},T}^i$, $n_{batch}$ is the number of elements in the batch, and $n_{seq}^i$ is the number of elements in the sequence of example $i$ in the batch. This objective differs from the traditional LLM fine-tuning objective of maximizing the average likelihood of the next token for each token in a batch of sequences as described in equation 2;

$$Loss = \frac{1}{\textcolor{red}{\sum_{i=1}^{n_{batch}} n_{seq}^i}} \sum_{i=1}^{n_{batch}} \left( -\log p_{\boldsymbol{\theta}}(d_{\mathcal{G},T}^{1,i}|\mathcal{D}_f^i) + \sum_{j=2}^{n_{seq}^i} -\log p_{\boldsymbol{\theta}}(d_{\mathcal{G},T}^{j,i}|d_{\mathcal{G},T}^{1:(j-1),i}, \mathcal{D}_f^i) \right) \quad (2)$$

The difference between the two objectives is the red term in both equations. We can view the reciprocal of this term as a weight placed on the importance of an example in the training set with respect to the training objective. For the standard training objective in equation 2, the term $\sum_{i=1}^{n_{batch}} n_{seq}^i$ in red changes from batch to batch, because batches contain randomly sampled examples which differ in sequence length. This means across batches, examples in the data set are weighted differently by the loss function. For the same reason, the loss of an example in the training set will be weighted differently across epochs based on the batch it falls in. Note that this is not an issue when training or fine-tuning on text data, because in most cases all training examples in a batch have the same length. A comparison in the performance of models trained with these objectives can be seen in Tables 2 and 3. The baseline `regen` only differs from our proposed model `SGG-LLM` without message passing in that it was trained using equation 2 instead of 1. We also report additional experiments shown Appendix D which suggests what is causing the difference in performance.

---

[1]We used GraphSAGE (15) in all experiments

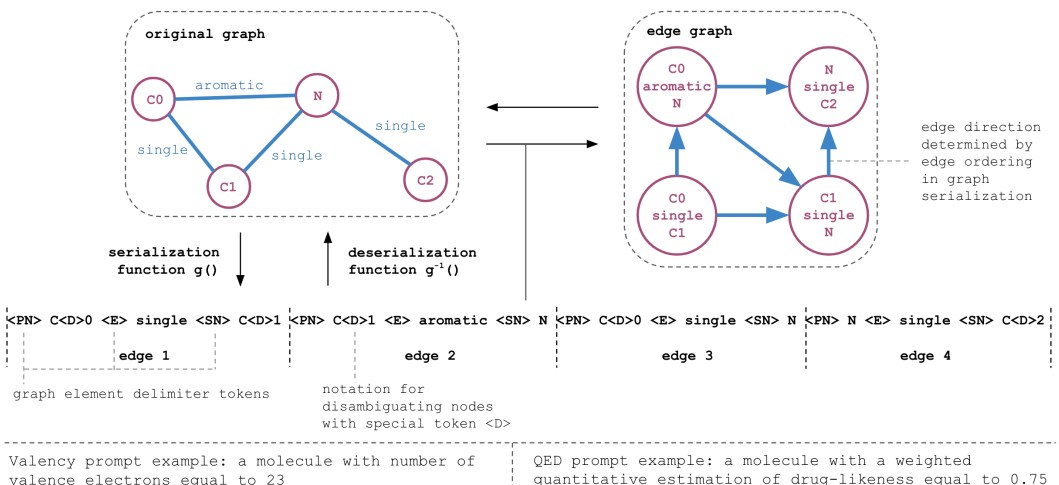

Figure 2: Top: the correspondence between a graph and its edge graph. Middle: the graph's `bag-of-edges` serialisation with special tokens `<PN>`, `<SN>`, `<E>` and `<D>`. Bottom: Examples of functional requirements used for condition graph generation for the molecule above.

## 3.2 NODE DISAMBIGUATION DURING SERIALIZATION

One special case of graph-structured data, which is not handled by prior work on generating serialized graphs, is when a subset of nodes in a graph are described by the same feature string[2]. However, this special case occurs in many domains, such as molecule data. For example, the molecule with multiple carbon atoms depicted in Figure 2 top row. To address this issue, we add a disambiguation token `<D>` to the feature strings of nodes with identical feature strings followed by a unique integer. An example of node disambiguation is shown the bottom row of Figure 2, where the three carbon atoms' feature strings are modified from `C, C, C` to `C<D>0, C<D>1, C<D>2`. With the addition of a disambiguation token, the serialization method becomes expressive enough to reversibly serialize any graph. In conjunction with this serialization function and special token `<D>`, we also add special tokens `<PN>` (predecessor node) `<E>` (edge) and `<SN>` (successor node) to the tokenizer of the pre-trained LLM, as well as additional rows to the embedding layer of the LLM for all four special tokens before fine-tuning. See our publicly available code for implementations of the proposed serialization $g(\cdot)$ and deserialization functions $g^{-1}(\cdot)$.

## 3.3 MESSAGE PASSING WITHIN A LANGUAGE MODEL

Our proposed architecture requires passing information from an LLM layer to an MP layer, then to the following LLM layer, and so on as shown in block B in Figure 1. As such, we need to convert the serialized graph representation outputted by an LLM layer into a representation of the graph that can be fed into an MP layer and vice versa. The inputs to an MP layer are node feature vectors $\mathcal{D}_{\mathcal{V},F} \in \mathbb{R}^{N \times H}$ for each node in a graph, where $H$ is the dimensionality of a feature vector, and the graph's adjacency matrix $\mathbf{A} \in \mathbb{R}^{N \times N}$. An obvious choice for constructing $\mathcal{D}_{\mathcal{V},F}$ would be simply using the feature vectors of $\mathcal{D}_{\mathcal{G},F}$. However, in $\mathcal{D}_{\mathcal{G},T}$, each node in the graph is described by multiple tokens and consequently multiple feature vectors in $\mathcal{D}_{\mathcal{G},F}$. For example, in Figure 2, the predecessor node of `edge 1` is described by four tokens [`<PN>`, `C`, `<D>`, `0`]. We can calculate a single node vector from its multiple token feature vectors in $\mathcal{D}_{\mathcal{G},F}$ by selecting the feature vector of the last element describing the node in $\mathcal{D}_{\mathcal{G},F}$ because the last vector contains information about all previous tokens in the sequence including all those describing the node because of the causal attention layers in an LLM.

---

[2]Prior work on generating graphs conditioned on text focused on tasks requiring graphs that did not contain sets of nodes with the same feature string (8; 27; 33)

In addition, if a node has a degree greater than one, it will be described more than once in $\mathcal{D}_{\mathcal{G}}$ [3]. While a node may occur more than once in a serialized graph, using our proposed serialization method, an edge will only occur once. In a graph $\mathcal{G} = \{\mathcal{V}, \mathcal{E}\}$, let there be a set $e^k$ for each edge $(v_i, v_j) \in \mathcal{E}$ where the superscript $k$ indicates the position of the edge in $\mathcal{D}_{\mathcal{G}}$ and the set contains the two nodes the edge connects $\{v_i, v_j\} \in e^k$. For the graph $\mathcal{G}$ and graph serialization $\mathcal{D}_{\mathcal{G}}$ pair depicted in Figure 2, $e^2$ would be the set of nodes $\{\texttt{C<D>1, N}\}$ contained in `edge 2` because `edge 2` is the second edge ($k = 2$) described in the serialization $\mathcal{D}_{\mathcal{G}}$. As a slight abuse of notation, we denote elements of a graph's edge set $\mathcal{E}$ either by $(v_i, v_j)$ or by $e^k$. Hence, we define a new graph $\mathcal{G}_{edge} = \{\mathcal{V}_{edge}, \mathcal{E}_{edge}\}$ where the edges in the original graph are treated as nodes $\mathcal{V}_{edge} = \mathcal{E}$ and nodes in the original graph define edges in the new graph $\mathcal{E}_{edge} = \{(e^j, e^k) | e^j, e^k \in \mathcal{E} \wedge e^j \cap e^k \neq \emptyset \wedge j \neq k\}$. We call this new graph an edge graph. Figure 2 shows an example of transforming a graph into its edge graph where the four edges in the original graph become the nodes in the edge graph. Let $f_{index} : \mathbb{Z}^+ \to \mathbb{Z}^+$ be a mapping from the index of an edge in the sequence $\mathcal{D}_{\mathcal{V}_{edge},F}$ to the index of the last token describing the edge in $\mathcal{D}_{\mathcal{G},F}$. To construct the sequence of feature vectors for the nodes in an edge graph $\mathcal{G}_{edge}$, for each MP layer we use the last token's feature vector of each edge in the serialized graph, so that $d^k_{\mathcal{V}_{edge},F} = d^{f_{index}(k)}_{\mathcal{G},F}$.

MP layers pass information between nodes in a graph by aggregating feature vectors from neighboring nodes. To ensure that an MP layer does not pass information backwards in $\mathcal{D}_{\mathcal{G},F}$, the MP layer should only aggregate information for a node from nodes that are earlier in the sequence as depicted in `causal graph` attention in block D in Figure 1. `causal` attention, shown in block C in Figure 1, only attends to elements previous to the query element in the sequence. `graph` attention only attends to elements in the sequence which are neighboring nodes to the query element in the sequence based on a reference graph. `causal graph` attention respects both of these constraints. We propose MP layers to pass information on a graph's edge graph. To ensure information is not passed backwards in $\mathcal{D}_{\mathcal{G},F}$ we must add an additional constraint when constructing the edge set of an edge graph which is $\forall (e^j, e^k) \in \mathcal{E}_{edge}, k > j$. This constraint has been applied to the edge graph shown in Figure 2. After passing an edge graph through an MP layer, we add each resultant edge feature vector to the token feature vector immediately after its description in the serialized graph as described in equation 3;

$$d^t_{\mathcal{G},F} = \begin{cases} d^t_{\mathcal{G},F} + d^k_{\mathcal{V}_{edge},F} \tanh a, & \text{if } f_{index}(k) = t - 1; \\ d^t_{\mathcal{G},F}, & \text{otherwise.} \end{cases} \tag{3}$$

During the development of this method, we found it necessary to multiply element-wise the output of an MP layer by a gating term $\tanh a$ when incorporating the output back into $\mathcal{D}_{\mathcal{G},F}$ as shown in equation 3. $a$ is a learned gating parameter of the MP layer initialized to $0$. Without the gating term, we could not fine-tune an LLM with message passing incorporated to generate valid graphs as demonstrated by the results shown in Appendix D. We hypothesize this is because the gating term allows an LLM to gradually learn to take into account graph structure. Without it, the model most likely starts fine-tuning at a position on the loss landscape from which it cannot converge to a useful local minima. The use of the gating term is inspired by the work of (1), which faced a similar issue.

## 4 RELATED WORK

### 4.1 INCORPORATING OTHER MODALITIES INTO TEXT GENERATION

This work proposes incorporating graph structure, an additional modality, into an LLM's text generation process (where the generated text is a serialized graph). There have been many works on incorporating other modalities into the text generation process. Speech-to-text can be posed as a sequence to sequence problem (16) and so similar methods to those used for text-to-text generation have been used to incorporate the modality of sound into language generation (2; 30; 37). Another widely studied problem is image-to-text generation, where most works focus on calculating dense sequence representations of images (21; 22; 24; 29; 40; 1) to pose image-to-text generation as a

---

[3]This makes it difficult to introduce graph information into the early elements in the graph serialization sequence. For a node with a degree greater than one, if we passed the feature vector outputted from an MP layer into its earliest instance in the graph serialization, we would be passing information backwards in the sequence

sequence-to-sequence generation problem as well. From this field of work, our proposed method is inspired by (1) which incorporates image information into the text generation process by representing sequences of images as sequences of dense vectors and interleaving special layers for combining the modalities of image and text between language modelling layers in an LLM. Inspired by this approach, we propose interleaving additional layers in-between language modelling layers in a pretrained LLM and incorporating additional tokens for representing the other modality. Unlike (1), we incorporate the modality of graphs as opposed to images into the generation process, and use a modified LLM to generate graphs rather than text.

## 4.2 GENERATING GRAPHS CONDITIONED ON TEXT

In the field of natural language processing, there have been many works on parsing natural language into syntax trees, some of which pose the problem as a serialized graph generation conditioned on text (9; 36). However, these works use a domain specific serialization method and do not incorporate graph structure into their proposed models' generation processes. There is one recent work (33) focused on generating explanation graphs from text which proposes a contrastive learning objective for fine-tuning the LLM T5 (31) to generate graphs, however the contrastive objective requires additional labels created by an expert. There is also a recent work published on generating protein graphs conditioned on text (25), however it cannot generalize past the domain of protein graph generation because the proposed method generates sequences of characters describing amino acid sequences.

Most prior work on text to graph generation focuses on generating knowledge graphs from text (8; 12; 14; 26; 27; 38). The best performing method on benchmark tasks is regen (8), which uses the LLM T5 (31) to generate knowledge graphs and is fine-tuned using the standard objective of predicting the next token in a sequence (equation 2). The second best performing method on benchmark knowledge graph generation tasks is grapher (27) which uses a combination of T5 and a recurrent neural network to generate graphs. This method differs from ours in the use of an additional model besides the LLM to predict edges and generate edge features as well as a training objective similar to equation 2.

## 5 EXPERIMENTS

Prior work on text graph generation conditioned on text does not provide an experimental design for evaluating methods that generate graphs based on functional descriptions. To do so, a dataset containing (functional requirements, graph) pairs is required for a task where it is possible to automatically calculate the functional properties of newly generated graphs. To generate such a dataset, we took the open graph benchmark large scale molecule dataset PCQM4M (20) and used the open source Python package rdkit (13) to generate two functional descriptions for each molecule: number of valency electrons in it and quantitative estimated-likeness (QED) (3). These are commonly used functional properties of a molecule's graph and are described in more detail in Appendix C.

### 5.1 DATASETS

For evaluation we created two datasets, each dataset is composed of all the molecules in the original PCQM4M (20) dataset with less than 20 atoms, which is still more than 2 million examples. One of the two datasets contains (number of valency electrons functional description, graph) pairs and is called Valency dataset. The other dataset contains (QED functional description, graph) pairs and is called QED dataset. To evaluate the impact of amount of data on model performance, we created three subsets for each of the datasets. Each subset had 1000 randomly selected molecules in its test and validation sets. Importantly, these were the same across the three subsets and then the training sets of the three subsets were another $25,000$, $100,000$ and $400,000$ randomly chosen examples. The training set of the smaller subsets was contained in the training set of the larger subset sets. See Figure 2 for examples of input output pairs for the constructed data sets.

|  | Minimum | Maximum | Mean | Standard Deviation | Median | Interquantile Range |
|---|---|---|---|---|---|---|
| Number of Valency Electrons | 2 | 122 | 77.3 | 13.3 | 80 | 70 - 88 |
| QED | 0.06 | 0.98 | 0.764 | 0.133 | 0.78 | 0.68 - 0.88 |

Table 1: Summary statistics about functional requirements in datasets used in experiments. The standard deviations reported for both data sets provide context for evaluating Table 2 below.

| Model | Message Passing | Training Set Size | Mean Absolute Error | |
|---|---|---|---|---|
|  |  |  | QED | Valency |
| SGG-LLM | none | 100,000 | $0.044 \pm 0.011$ | $0.060 \pm 0.018$ |
| SGG-LLM | edges | 100,000 | $0.036 \pm 0.005$ | $0.035 \pm 0.014$ |
| SGG-LLM | correspondences | 100,000 | $0.039 \pm 0.007$ | $0.045 \pm 0.017$ |
| SGG-LLM | none | 25,000 | $0.062 \pm 0.008$ | $1.703 \pm 0.074$ |
| SGG-LLM | none | 400,000 | $0.020 \pm 0.001$ | $0.076 \pm 0.034$ |
| grapher | N/A | 100,000 | $0.157 \pm 0.004$ | $1.268 \pm 0.229$ |
| regen | N/A | 100,000 | $0.149 \pm 0.018$ | $2.282 \pm 1.156$ |

Table 2: Model performance in terms of MAE on QED and Valency datasets. The first, second, and third best performing models are highlighted using the colors shown here. Experiments were repeated three times to estimate standard error. All fine-tuned variants of SGG-LLM outperform baselines by a statistically significant margin. Models below the boldface line are from prior work.

## 5.2 METRICS AND EVALUATION

To evaluate models, we calculated three metrics on the test sets of the datasets described above: parsability, diversity, and most importantly mean absolute error (MAE) with respect to the conditional functional property. A generated graph is parsable if no error is thrown when calculating its functional property. If a molecule is parsable (i.e. it follows correct serialization syntax and doesn't violate basic laws of physics), it is given a score of 1, otherwise 0. In all experiments we use the following metrics:

- *Parsability* is the mean parsability score of samples in the test set.

- *MAE* is the mean absolute error between the generated graph's functional property value and the requested property value averaged over samples in the test set. To interpret the magnitudes reported for MAE results see Table 1 describing some summary statistics about functional properties of graphs in the datasets used in experiments.

- *Diversity* is a measure of the multimodality of the distribution $p_\theta(\mathcal{D}_\mathcal{G}|\mathcal{D}_f)$ that the LLM learns. A sampled graph is assigned a diversity score of 1, if it, a second sampled graph and the ground truth do not share the same node and edge sets; and is assigned a score of 0 otherwise. Diversity is the average of diversity scores of samples in the test set.

## 5.3 CURRENT STATE-OF-THE-ART

We implemented current state-of-the-art models from prior work as baselines: specifically we implemented the version of grapher (27), which generates edge features instead of classifying them and regen (8). We had to make two modifications to both approaches so they could generalize to molecule data: we added node disambiguation from Section 3.2 to their serialization methods and updated their language models to a more recent model BLOOM (34), which is the same LLM used in experiments by our proposed approach. See Appendix A for a description of the training process used to train grapher, regen and variants of our proposed approach.

## 6 RESULTS

On the task of generating graphs to meet functional requirements, the ideal model can generate a diverse set of parsable graphs with functional properties equal to the requested functional property. Tables 2 and 3 describe the results of our proposed approach on the QED and Valency datasets. Our

| Model | Message Passing | Training Set Size | Parsability | | Diversity | |
|---|---|---|---|---|---|---|
| | | | QED | Valency | QED | Valency |
| SGG-LLM | none | 100,000 | 1.000 ± 0.000 | 0.999 ± 0.001 | 0.845 ± 0.017 | 0.506 ± 0.031 |
| SGG-LLM | edges | 100,000 | 0.998 ± 0.001 | 0.999 ± 0.000 | 0.836 ± 0.029 | 0.540 ± 0.016 |
| SGG-LLM | correspondences | 100,000 | 0.995 ± 0.001 | 0.998 ± 0.000 | 0.839 ± 0.008 | 0.608 ± 0.054 |
| SGG-LLM | none | 25,000 | 0.997 ± 0.002 | 0.991 ± 0.003 | 0.799 ± 0.008 | 0.518 ± 0.078 |
| SGG-LLM | none | 400,000 | 0.998 ± 0.001 | 1.000 ± 0.000 | 0.857 ± 0.006 | 0.542 ± 0.051 |
| grapher | N/A | 100,000 | 1.000 ± 0.000 | 1.000 ± 0.000 | 0.745 ± 0.038 | 0.410 ± 0.045 |
| regen | N/A | 100,000 | 0.984 ± 0.008 | 0.991 ± 0.007 | 0.854 ± 0.031 | 0.446 ± 0.124 |

Table 3: Model performance in terms of parsability and diversity on QED and Valency datasets. The first, second, and third best performing models are highlighted using the colors shown here. Experiments were repeated three times to estimate standard error. Models below the boldface line are from prior work. All models achieved a parsability near 1.0.

proposed approach is referred to as SGG-LLM standing for serialized graph generator large language model. All variants of our approach and the baselines grapher, and regen achieve a parsability score near or at 1. The SGG-LLM variant with correspondences message passing (described in Appendix B) is another method of incorporating the MP layers into an LLM by passing messages in $\mathcal{D}_\mathcal{G}$ based on node correspondences rather than edges.

Results in table 2 suggest that all variants of SGG-LLM outperform baselines by a statistically significant margin, using the unpaired t-student test and a threshold p-value of $0.05$, in terms of generating examples with functional properties close those requested. In addition, results suggest that all variants of SGG-LLM outperform grapher by a statistically significant margin in terms of generating a diverse set of candidates on the Valency dataset. Finally, the two variants of SGG-LLM that use MP layers outperform the variant that does not utilise MP layers.

The high performance of all variants of our approach over regen suggest the importance of the proposed loss function and of weighting tokens equally across batches during training. The high performance of all variants of our approach over grapher could be for the same reasons as regen, as it is trained with a similar loss, or it could be that single model which jointly estimates the existence of nodes, edges, and their features is more effective at generating graphs than a dual module model.

To demonstrate the generality of our proposed approach beyond generating molecules, we also evaluate it on the benchmark knowledge graph generation dataset WebNLG+ 2020 (10) using a different language model T5 (31). Note this task is generating graphs conditioned on an imperative description of the graph, so is not directly linked to the focus of this paper. See Appendix E for a description of the experiments and discussion of the results.

# 7 DISCUSSION

The main limitation of our proposed approach is its reliance on fine-tuning a pre-trained autoregressive LLMs, which are computationally expensive, slow at generation time and require substantial computational resources even to fine-tune. This limitation would become even more difficult when applying this method to tasks containing larger graphs or graphs containing elements with long feature strings. Hence an interesting next step from this method would be using quantized pre-trained models and low rank adapters on LLM layers for more efficient fine tuning (7).

In terms of societal impact, our proposed approach might help speed up and improve processes such as drug discovery, software development and project planning. But at the same time requires oversight to ensure it is not used for nefarious applications like the design of chemical weapons. In addition, if this approach is applied to a task in the social sciences, analysis should be required to ensure that the biases learned by the model are understood and any unfair preferences learned for a certain demographic or group should be mitigated.

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

## A  HYPERPARAMETERS AND TRAINING

For experiments on the molecule data sets, all variants of our proposed approach and the implemented baseline `grapher` (27) and `regen` (8) used the pretrained version of `BLOOM` (34) with 560 million parameters as their language model. For experiments on the WebNLG2020+ data set (10), all models used the pretrained version of `T5` (31) with 770 million parameters as their language model. All models were trained for up to ten epochs and checkpointed based on minimizing validation loss. Model's were trained using the ADAM optimizer (23) with a learning rate of $3e - 5$, a $\beta_1$ of 0.9, a $\beta_2$ of 0.999 and a regularization weight of $1e - 7$ as well as a linear learning rate schedule. During training, model parameters' gradients norms were clipped to a value of 1.0. The models were trained with a batch size of 18 using stage 2 data parallelism using the method described in (32). All models were trained and evaluated on machines with 3 NVIDIA A100 GPUs. Variants of our proposed approach which incorporated message passing into the LLM by interleaving message passing layers in the LLM used a single GraphSage (15) layer in each message passing layer with an embedding size equal to the token embedding size of the LLM they were incorporated into.

## B  MESSAGE PASSING BETWEEN NODE CORRESPONDENCES

Most nodes appear at least twice in a `bag-of-edges` serialization. We can define a correspondence graph from a serialized graph $\mathcal{D}_\mathcal{G}$ by treating each instance of a node in a serialized graph $\mathcal{D}_\mathcal{G}$ as its own node in the correspondence graph. Then the correspondences between instances define edges in the correspondence graphs. An example of a correspondence graph is shown in the graphic below;

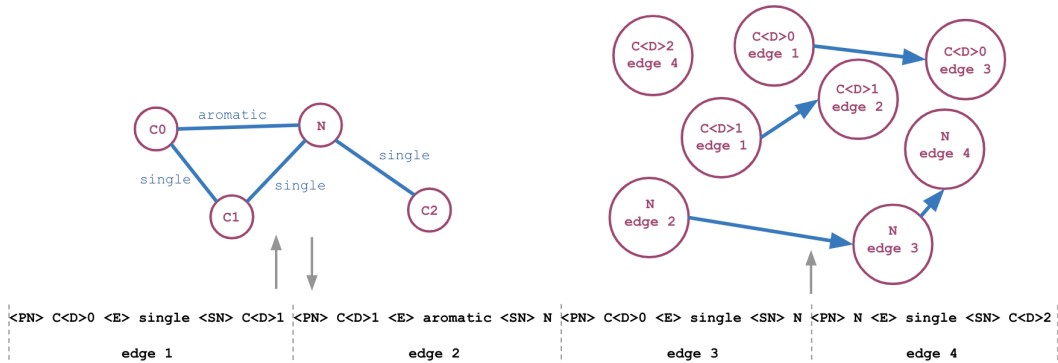

In the graphic, the node `C<D>0` occurs more than once in $\mathcal{D}_\mathcal{G}$ and each instance is treated as its own node in the correspondence graph, so the node `C<D>0` corresponds to `C<D>0 edge 1` and to `C<D>0 edge 3` in the correspondence graph. Edges are constructed by connecting nodes in the correspondence graph which correspond to the same node in the original graph i.e. in the graphic `C<D>0 edge 1` is connected to `C<D>0 edge 3` because they refer to the node `C<D>0` in the original graph. Nodes in the correspondence graph are only connected if they are adjacent occurrences in the serialized graph. By constructing edges such that they always point from an earlier instance of a node to a later instance we ensure the message passing layer does not pass information backwards in the serialized graph sequence at training time. As additional method, we propose incorporating MP layers into an LLM where the MP layers pass information based on a graph's correspondence graph.

## C  FUNCTIONAL DESCRIPTIONS OF MOLECULES

We used two functional descriptions of molecules to generate datasets for our experiments - number of valency electrons and QED. The number of valency electrons in a molecule is the sum of the number of valency electrons in its atoms. Importantly, this property is a result of only a graph's node composition, but not its full structure. To generate descriptions of a functional property of a graph's full structure, we calculate a metric called quantitative estimated drug-likeness (QED) (3). QED quantifies multiple properties of a molecule which are attractive for applications in drug design and then calculates a single metric from these properties using a weighted logarithmic sum. The properties used to calculate QED are molecular weight, octanol–water partition coefficient, topological polar surface area, number of hydrogen bond donors and acceptors, the number of aromatic rings and rotatable bonds, and the presence of unwanted chemical functionalities. For experiments we set the weight of the properties: molecular weight, number of hydrogen bond donors and acceptors, and presence of unwanted chemical functionalities to zero in the QED calculation, because they are mainly determined by node composition instead of the entire graph structure. The functional descriptions generated were of the format *"a molecule with number of valence electrons equal to ..."* and *"a molecule with a weighted quantitative estimation of drug-likeness equal to ..."*.

## D  ADDITIONAL RESULTS OF MOLECULE GENERATION EXPERIMENTS

Below, we provide three tables describing the full results of experiments on the molecule data sets including ablation studies to empirically justify some design choices in our proposed method. Our proposed model is referred to as `SGG-LLM`. There are three additional models in the tables below; 1) `SGG-LLM` w/out fine-tuning, 2) `SGG-LLM` w/ special loss, and 3) `SGG-LLM` with edge based message passing without a gating term. The table below describing performance of models in terms of the parsability of generated examples shows that `SGG-LLM` w/out fine-tuning and `SGG-LLM` with edge based message passing without a gating term both did not produce a single parsable example on molecule datasets' test sets. These results suggests that both fine-tuning and a gating term (when incorporating message passing into an LLM) are required to achieve good performance with our proposed method. Note if a model cannot generate parsable examples, then the metric mean absolute error cannot be calculated and the diversity of generated examples is not a useful measure of model performance. Consequently, we do not report mean absolute error or diversity for `SGG-LLM` w/out fine-tuning and `SGG-LLM` with edge based message passing without a gating term.

| Model | Message Passing | Training Set Size | Parsability | |
| --- | --- | --- | --- | --- |
| | | | QED | Valency |
| `SGG-LLM` | none | 100,000 | $1.000 \pm 0.000$ | $0.999 \pm 0.001$ |
| `SGG-LLM` w/out fine-tuning | none | 100,000 | $0.000 \pm 0.000$ | $0.000 \pm 0.000$ |
| `SGG-LLM` w/ special loss | none | 100,000 | $0.986 \pm 0.003$ | - |
| `SGG-LLM` | edges | 100,000 | $0.998 \pm 0.001$ | $0.999 \pm 0.000$ |
| `SGG-LLM` w/out gating term | edges | 100,000 | $0.000 \pm 0.000$ | $0.000 \pm 0.000$ |
| `SGG-LLM` | correspondences | 100,000 | $0.995 \pm 0.001$ | $0.998 \pm 0.000$ |
| `SGG-LLM` | none | 25,000 | $0.997 \pm 0.002$ | $0.991 \pm 0.003$ |
| `SGG-LLM` | none | 400,000 | $0.998 \pm 0.001$ | $1.000 \pm 0.000$ |
| `grapher` | N/A | 100,000 | $1.000 \pm 0.000$ | $1.000 \pm 0.000$ |
| `regen` | N/A | 100,000 | $0.984 \pm 0.008$ | $0.991 \pm 0.007$ |

The model `SGG-LLM` w/ special loss is a version of `SGG-LLM` without message passing that was trained using equation 1, but instead of letting $n_{batch}$ = batch size as proposed in section 3.1, we equal $n_{batch} = E[\sum_{i=1}^{n_{batch}} n_{seq}^i]$ which is the expected value of the differing term in equation 2 from equation 1. This is to determine whether it is the magnitude of the differing term causing the difference in performance or the fact that $\sum_{i=1}^{n_{batch}} n_{seq}^i$ changes from batch to batch. In experiments, some summary statistics for the term $\sum_{i=1}^{n_{batch}} n_{seq}^i$ in equation 2 were mean = 4053, max = 4676, median = 4060, and inter-quartile range = 3915 − 4199. There was a marginal difference in performance in terms of all three metrics when comparing the performance of `SGG-LLM` w/ special

loss and `SGG-LLM` without message passing. The marginal difference in performance suggests that the changing weighting between batches and across epochs is what hurts the performance of models trained with the objective described in equation 1. `regen` is a model trained with equation 1.

| Model | Message Passing | Training Set Size | Mean Absolute Error | |
|---|---|---|---|---|
| | | | QED | Valency |
| `SGG-LLM` | none | 100,000 | $0.044 \pm 0.011$ | $0.060 \pm 0.018$ |
| `SGG-LLM` w/ special loss | none | 100,000 | $0.046 \pm 0.004$ | - |
| `SGG-LLM` | edges | 100,000 | $0.036 \pm 0.005$ | $0.035 \pm 0.014$ |
| `SGG-LLM` | correspondences | 100,000 | $0.039 \pm 0.007$ | $0.045 \pm 0.017$ |
| `SGG-LLM` | none | 25,000 | $0.062 \pm 0.008$ | $1.703 \pm 0.074$ |
| `SGG-LLM` | none | 400,000 | $0.020 \pm 0.001$ | $0.076 \pm 0.034$ |
| `grapher` | N/A | 100,000 | $0.157 \pm 0.004$ | $1.268 \pm 0.229$ |
| `regen` | N/A | 100,000 | $0.149 \pm 0.018$ | $2.282 \pm 1.156$ |

| Model | Message Passing | Training Set Size | Diversity | |
|---|---|---|---|---|
| | | | QED | Valency |
| `SGG-LLM` | none | 100,000 | $0.845 \pm 0.017$ | $0.506 \pm 0.031$ |
| `SGG-LLM` w/ special loss | none | 100,000 | $0.831 \pm 0.003$ | - |
| `SGG-LLM` | edges | 100,000 | $0.836 \pm 0.029$ | $0.540 \pm 0.016$ |
| `SGG-LLM` | correspondences | 100,000 | $0.839 \pm 0.008$ | $0.608 \pm 0.054$ |
| `SGG-LLM` | none | 25,000 | $0.799 \pm 0.008$ | $0.518 \pm 0.078$ |
| `SGG-LLM` | none | 400,000 | $0.857 \pm 0.006$ | $0.542 \pm 0.051$ |
| `grapher` | N/A | 100,000 | $0.745 \pm 0.038$ | $0.410 \pm 0.045$ |
| `regen` | N/A | 100,000 | $0.854 \pm 0.031$ | $0.446 \pm 0.124$ |

# E  RESULTS OF KNOWLEDGE GRAPH GENERATION EXPERIMENTS

To demonstrate the generality of our proposed approach beyond generating molecules, we also evaluate it on the benchmark knowledge graph generation dataset WebNLG+ 2020 (10) using a different language model `T5` (31). Like in prior work, model performance is evaluated using the metrics of F1-score, precision and recall when comparing a generated graph to the ground truth knowledge graph. See (8) for a more detailed explanation of these metrics. We compare our proposed method to the three best performing models on this data set; `regen` (8), `grapher` (27), and `bt5` (11).

On the WebNLG+ 2020 data set, results in the table below suggest that the incorporating message passing into an LLM is not useful to the task of knowledge graph generation from an imperative description, but also that using edge message passing does not degrade performance. Interestingly, `SGG-LLM` without message passing was able to achieve state-of-the-art performance on the benchmark task of generating triples in a knowledge graph. `regen`'s implementation for experiments on WebNLG2020+ was identical to `SGG-LLM` without message passing, except that `SGG-LLM` was trained with a different training objective (see equation 1 in the main paper), a more aggressive learning rate and a linear learning rate schedule. So the state-of-the-art performance of `SG-LLM` on WebNLG2020+ may be attributed to better hyperparameter selection or the modified training objective.

| Model | Exact | | | Partial | | | Strict | | |
|---|---|---|---|---|---|---|---|---|---|
| | F1 | Prec. | Rec. | F1 | Prec. | Rec. | F1 | Prec. | Rec. |
| `bt5` | 0.682 | 0.670 | 0.701 | 0.713 | 0.700 | 0.736 | 0.675 | 0.663 | 0.695 |
| `regen` | 0.723 | 0.714 | 0.738 | 0.767 | 0.755 | 0.788 | 0.720 | 0.713 | 0.735 |
| `grapher` | 0.683 | 0.675 | 0.695 | 0.713 | 0.702 | 0.730 | 0.681 | 0.673 | 0.693 |
| `SGG-LLM` none | 0.747 ±0.005 | 0.743 ±0.003 | 0.765 ±0.007 | 0.779 ±0.006 | 0.771 ±0.005 | 0.798 ±0.004 | 0.726 ±0.005 | 0.720 ±0.007 | 0.731 ±0.012 |
| `SGG-LLM` edges | 0.754 ±0.003 | 0.748 ±0.002 | 0.762 ±0.004 | 0.786 ±0.001 | 0.779 ±0.000 | 0.795 ±0.003 | 0.733 ±0.005 | 0.729 ±0.004 | 0.741 ±0.003 |
| `SGG-LLM` correspo-ndences | 0.743 ±0.008 | 0.738 ±0.009 | 0.752 ±0.005 | 0.774 ±0.005 | 0.769 ±0.009 | 0.787 ±0.002 | 0.683 ±0.067 | 0.693 ±0.081 | 0.691 ±0.069 |

Model performance on knowledge graph data set WebNLG+ 2020. Note: baseline model results do not report standard deviations because they were not reported in prior work and we felt it was more appropriate to report baseline results based their published results as opposed to reimplementing the baselines ourselves. Experiments with variants of our proposed approach were repeated three times to estimate standard error.

