# OpenReview forum: "Form follows Function: Text-to-Text Conditional Graph Generation based on Functional Requirements"
_ICLR.cc/2024/Conference — Submitted to ICLR 2024_

### Official Review · Reviewer_Q4ZR · 2023-10-28

**Soundness:** 2 fair
**Presentation:** 2 fair
**Contribution:** 3 good
**Rating:** 5
**Confidence:** 4

**Summary:**

The paper proposes a new graph generation framework based on pretrained LLM. The paper formulates the graph generation problem as the generation of a serialized graph. The paper first proposes a serialization method that reversibly serializes any graph by converting the node graph into an edge graph. The paper also slightly changes the weight of NLL loss to ensure each instance is weighted equally. The paper finally proposes a new message-passing layer above the language model. The paper conducts experiments on molecular and WdbNLG graph generation and achieves competitive results.

**Strengths:**

1. The paper proposes a new graph generation technique based on pretrained LLMs. The paper first modifies the weight of the NLL loss. The paper then inserts a message-passing layer into the standard transformers. The paper finally proposes a new edge graph to reduce node ambiguity.
2. The paper conducts experiments on molecular generation for QED and valency. The proposed framework achieves significant improvements compared to other baselines. The paper further analyzes its parsability and diversity. Additionally, the paper also conducts experiments on the WebNLG tasks to show its generalization ability. The paper includes an additional ablation study in the Appendix.
3. The paper provides code. The paper provides comprehensive implementation details in the Appendix.

**Weaknesses:**

1. The idea of message passing for molecular graphs is incremental. For example, Klicpera et al., show message passing is important for molecular graphs, although their message-passing function is different from this paper. The idea of fine-tuning objectives is also an engineering trick rather than a model contribution.
2. BLOOM is not a suitable baseline for molecular generation since it is not trained on molecular data. Galactica (Taylor et al., 2022) would be better since its training data includes smile strings. The simple autoregressive model can outperform the proposed message passing by increasing its training set.
3. The language in the paper needs to be further polished.



Klicpera, J., Groß, J., & Günnemann, S. (2003). Directional Message Passing for Molecular Graphs 2020. ICLR 2020 https://arxiv.org/pdf/2003.03123.pdf

Taylor, R., Kardas, M., Cucurull, G., Scialom, T., Hartshorn, A., Saravia, E., ... & Stojnic, R. (2022). Galactica: A large language model for science. arXiv preprint arXiv:2211.09085. https://galactica.org/static/paper.pdf

**Questions:**

Is it possible to include additional analysis for Table 3, since sometimes baselines perform better?

---

> ### Author Response · Authors · 2023-11-16
>
> [Reviewer] The idea of message passing for molecular graphs is incremental. For example, Klicpera et al., show message passing is important for molecular graphs, although their message-passing function is different from this paper. The idea of fine-tuning objectives is also an engineering trick rather than a model contribution.
>
> [Authors] It seems that we could improve the way we described our technical contributions in the paper. Our goal was to propose using an LLM to generate graphs and interleaving message passing layers in between LLM layers to improve the LLMs performance. To the best of our knowledge this is a novel proposal. We propose fine tuning as part of our method as we found it is the only way to get a model to output parsable serialized graphs, not because we view it as a technical contribution. The contribution with respect to fine-tuning in our opinion is the novel training objective we propose which is not an engineering trick by a modification of the standard autoregressive training objective that takes into account the structure of our data. How could we make our technical contributions clearer so that it does not seem like we are proposing message passing for molecule graphs?

---

> ### Author Response · Authors · 2023-11-16
>
> [Reviewer] BLOOM is not a suitable baseline for molecular generation since it is not trained on molecular data. Galactica (Taylor et al., 2022) would be better since its training data includes smile strings.
>
> [Authors] While the experiments for this paper focus on the application of molecule generation, the focus of the paper itself is not a new method for performing molecule generation. The focus is a new method for generating graphs through serialized graph generation that match conditional functional requirements. Why do you feel that BLOOM as a language model is not suitable for the general problem of generating graphs through serialized graph generation that match conditional functional requirements?

---

> ### Author Response · Authors · 2023-11-16
>
> [Reviewer] The simple autoregressive model can outperform the proposed message passing by increasing its training set.
>
> [Authors] In many cases, the more data provided to a model, the less inductive bias (i.e. adding MP into an LLM) required to improve performance. But in most applications of data driven modelling, the amount of data available is a limiting factor. So even if two methods are perform comparably with enough data, the method that is more data efficient is preferred because it can be used in more real world situations. Given that experiments suggests our proposed method with MP is more efficient, we view the proposed method with MP as a contribution even if it has comparable performance to a model without MP with more data.

---

> ### Author Response · Authors · 2023-11-16
>
> The language in the paper needs to be further polished.
>
> Given this comment, are there are any specific areas of the paper you would like us to focus on? Or possibly a specific way we explain things that you think could be improved?

---

> ### Author Response · Authors · 2023-11-16
>
> [Reviewer] Is it possible to include additional analysis for Table 3, since sometimes baselines perform better?
>
> [Authors] What additional analysis would you like to see in Table 3? Although sometimes baselines perform better in Table 3, our main focus is on generating examples which meet functional requirements. The results in table 2 suggest that our proposed approaches always outperform baselines in terms of generating examples which more closely meet functional requirements.

---

> > ### Comment · Reviewer_Q4ZR · 2023-11-22
> >
> > Thank you for your response to my questions. The authors have answered most of my questions. However, I still believe the technical contribution is weak, as outlined in my review. In addition, the authors fail to provide additional experiments or update the paper content. Therefore, I maintain my original scores.

---

### Official Review · Reviewer_cSty · 2023-10-31

**Soundness:** 2 fair
**Presentation:** 2 fair
**Contribution:** 2 fair
**Rating:** 5
**Confidence:** 3

**Summary:**

The paper studies text-to-graph generation with large language models. The authors propose a new fine-tuning method in which LM fine-tuning and message-passing (MP) are interleaved. Empirical results indicate the effectiveness of the approach.

The problem in question is interesting, but I have some concerns:

(1) the technical contribution is not clear to me.  It seems that interleaving with MP is the only technical contribution, however, when the number of fine-tuning examples increases, the model without MP can achieve much better performance (half MAE on QED), then what will happen if we fine-tune the model on 400k examples with MP? is it possible that w/ MP and w/o MP are comparable when there are enough fine-tuning examples? Moreover, why the results on QED and Valency are inconsistent on 400k examples?

(2) the presentation needs improvement. I find conflict in some descriptions. For example, in the third line from the bottom in Page 4, it is said that "regen only differs from our proposed model SGG-LLM without message passing in that it was trained using equation 2 instead of 1" but in the third line from the top in Page 16, it is said that "regen is a model trained with equuation 1". I am confused with such presentations. Then what is the true difference between the proposed method and the baseline? It is also important to highlight the major contributions in Introduction.

**Strengths:**

interesting problem

**Weaknesses:**

presentation
insufficient empirical analysis

**Questions:**

see the summary

---

> ### Author Response · Authors · 2023-11-16
>
> [Reviewer] (1) the technical contribution is not clear to me.
>
> [Authors] We attempted to highlight the technical contribution of the paper in the introduction with the bold-faced sentence. In more detail, we believe there are four technical contributions;
> - A novel training objective that significantly improves performance of models on the proposed task
> - A novel serialization method that can reversibly serialize any graph
> - Interleaving MP layers in between LLM layers so that an LLM can take into account graph structure during generation
> - An experimental design to evaluate a model's ability to generate graphs that meets requested functional requirements
>
> Do you agree with these contributions? If yes, how do you think we could highlight them more clearly? If not, which do you think is not a contribution?

---

> ### Author Response · Authors · 2023-11-16
>
> [Reviewer] It seems that interleaving with MP is the only technical contribution, however, when the number of fine-tuning examples increases, the model without MP can achieve much better performance (half MAE on QED), then what will happen if we fine-tune the model on 400k examples with MP? is it possible that w/ MP and w/o MP are comparable when there are enough fine-tuning examples?
>
> [Authors] In many cases, the more data provided to a model, the less inductive bias (i.e. adding MP into an LLM) required to improve performance. But in most applications of data driven modelling, the amount of data available is a limiting factor. So even if two methods are perform comparably with enough data, the method that is more data efficient is preferred because it can be used in more real world situations. Given that experiments suggests our proposed method with MP is more efficient, we view the proposed method with MP as a contribution even if it has comparable performance to a model without MP with more data.

---

> ### Author Response · Authors · 2023-11-16
>
> [Reviewer] Moreover, why the results on QED and Valency are inconsistent on 400k examples?
>
> [Authors] They are inconsistent because they are not the same data set, the two data sets have different labels (QED vs Valency) which are not equally easy to learn to conditionally generate good examples from. Given that they are different data sets, we cannot be sure whether we should expect consistency between the two of them. Would it make the paper clearer if we separate the results of the two different data sets into different tables?

---

### Official Review · Reviewer_fDhs · 2023-11-09

**Soundness:** 2 fair
**Presentation:** 2 fair
**Contribution:** 2 fair
**Rating:** 3
**Confidence:** 4

**Summary:**

This paper proposes a new problem of graph generation based on given graph properties. The authors propose to use LLMs for graph generation. They add graph message passing layers into LLMs for capturing graph features based on graph structure.

**Strengths:**

* This paper proposes some techniques to solve the ambiguous nodes and incorporate graph structures when using LLMs for graph generation.

**Weaknesses:**

* The task appears to be ill-defined. The authors claim to introduce a "novel problem setting," but fail to provide a clear problem definition. It remains unclear what the key challenges of the proposed property-to-graph generation task are. Additionally, the evaluation dataset and metrics lack soundness discussion. If the evaluation benchmark can be derived from previous works with little modification, why not using the same benchmark and metrics in previous works? The authors should thoroughly address the suitability of the datasets and metrics. Baselines are also inadequately explored, and the results are far from convincing.
* The technical contribution is also limited. The primary contribution emphasized by the authors involves the use of causal masks during generation. However, the employment of causal masks for autoregressive generation is just a trick to ensure efficient training. (Otherwise, it will require to train on each token independently with multiple passes.) It's weird to label this well-known technical detail as the key contribution.
* The paper is hard to follow.

**Questions:**

* See weaknesses.

---

> ### Author Response · Authors · 2023-11-16
>
> [Reviewer] The technical contribution is also limited. The primary contribution emphasized by the authors involves the use of causal masks during generation. However, the employment of causal masks for autoregressive generation is just a trick to ensure efficient training. It's weird to label this well-known technical detail as the key contribution.
>
> [Authors] We politely disagree with the claim that causal masking is just a trick to ensure efficient training. In fact the time complexity of causal masking is the same as no masking. In addition, a model trained without causal masking will perform very poorly when generating text because without masking the model has access to the next token it is trying to predict during training, so it does not learn anything. Novel attention masking methods have been published in ICLR, ICML and NeurIPS before, so masking methods are considered a contribution. The masking method we are proposing allows the model to incorporate graph structure into the inference process which is a contribution because to the best of our knowledge it has not been proposed before.

---

> > ### Author Response · Authors · 2023-11-16
> >
> > [Reviewer] It remains unclear what the key challenges of the proposed property-to-graph generation task are.
> >
> > [Authors] The challenge of property-to-graph generation that we highlighted in the paper is that in most applications there will be multiple graphs that meet the same property, so it is difficult to evaluate a model’s performance on this task without the right experiment design. We state this in paragraph 2 of the introduction and in paragraph 1 of the preliminaries section. Would it improve the writing if we were explicit in highlighting this as a challenge?

---

> ### Author Response · Authors · 2023-11-16
>
> [Reviewer] The task appears to be ill-defined. The authors claim to introduce a "novel problem setting," but fail to provide a clear problem definition. It remains unclear what the key challenges of the proposed property-to-graph generation task are. Additionally, the evaluation dataset and metrics lack soundness, as they are derived from existing ones that were not originally designed for the newly proposed task. The authors should thoroughly address the suitability of the datasets and metrics. Baselines are also inadequately explored, and the results are far from convincing.
>
> [Authors] Your comments suggest that there is a lot of room for us to improve our paper and we would like to work with you to learn how we can improve our work.

---

> ### Author Response · Authors · 2023-11-16
>
> [Reviewer] The authors claim to introduce a "novel problem setting," but fail to provide a clear problem definition.
>
> [Authors] Could you provide us with more detail about what is unclear in our problem definition?

---

> ### Author Response · Authors · 2023-11-16
>
> [Reviewer] Additionally, the evaluation dataset and metrics lack soundness, as they are derived from existing ones that were not originally designed for the newly proposed task. The authors should thoroughly address the suitability of the datasets and metrics.
>
> [Authors] The evaluation data sets were modified to fit the newly proposed task using an open source software that allows us to calculate functional properties of molecules. Could you provide us with more detail what elements of the modified data set or the metrics we employed were not well-suited to the task we were evaluating models on?

---

> ### Author Response · Authors · 2023-11-16
>
> [Reviewer] Baselines are also inadequately explored, and the results are far from convincing.
>
> [Authors] Can you also provide more detail about why you feel baselines are inadequately explored?

---

> ### Comment · Area_Chair_5NBq · 2023-11-21
>
> Dear authors,
>
> 1. I agree that masking methods are worth being published definitely; can you highlight the novelty of the proposed masking method in this paper? If it's causal masking as pointed out by the reviewer, then it's hard for readers to appreciate its novelty. Can you elaborate?
>
> 2. In order for the reviewers and myself to take a more detailed look, can you point to the place where the problem is defined in the paper?
>
> Best,
> AC

---

> > ### Comment · Area_Chair_5NBq · 2023-11-21
> >
> > Dear reviewer fDhs,
> >
> > Since you mentioned that "Baselines are also inadequately explored, and the results are far from convincing", can you expand your point a bit to help the authors improve their next version? E.g., what other baselines do you think they should try, and why are their results not convincing?
> >
> > Best,
> > AC

---

> ### Comment · Reviewer_fDhs · 2023-11-22
>
> Hi AC,
>
> Generating a molecular graph based on properties is not an entirely novel task. Previous works [1,2], while not converting all inputs into textual sequences, can still be applicable in this context. This paper only compares their approach with two T5-based knowledge graph generation methods. Comparing with methods not specifically designed for this domain raises concerns about the persuasiveness of the comparison.
>
> [1] Jin, Wengong, Regina Barzilay, and Tommi Jaakkola. "Junction tree variational autoencoder for molecular graph generation." International conference on machine learning. PMLR, 2018.
> [2] Popova, Mariya, et al. "MolecularRNN: Generating realistic molecular graphs with optimized properties." arXiv preprint arXiv:1905.13372 (2019).
>
> Best, reviewer fDhs

---

> > ### Author Response · Authors · 2023-11-22
> >
> > Dear reviewer fDhs,
> >
> > Thank you for your response. Molecule generation conditioned on functional properties is the task we use to evaluate our model, but that is not the focus of this paper. Instead we focus on the task of generating graphs from functional requirements and focus on the method where an LLM is used as the model to generate serialized graph. We compare our proposed approach to prior work which focuses on the same general method. Our claim is that when using an LLM to generate serialized graphs based on functional requirements our proposed method is state-of-the-art, not that our proposed approach is state of the art for generating molecules based on properties. Taking this into account, incorporating the two methods you cite as possible additional baselines in our experiments would be adding content to the paper which would be providing evidence for an additional claim, it would not support the main claim this paper makes.
> >
> > Best, Authors

---

### Official Review · Reviewer_kFQj · 2023-11-09

**Soundness:** 2 fair
**Presentation:** 4 excellent
**Contribution:** 3 good
**Rating:** 5
**Confidence:** 4

**Summary:**

This paper introduces the problem of generating graphs conditioned on its functional requirements as a text-to-text generation task. Experiments were conducted on a novel formulation of PCQM4M for text graph generation, as well as WebNLG+ 2020. The proposed approach involves graph serialization / de-serialization that handles node disambiguation, a variant of the negative log-likelihood objective, and interleaving message passing layers between transformer layers to pass information via a graph's edge graph (or variants) with causal graph attention masking.

**Strengths:**

The paper is written clearly and is mostly easy to follow. The diagrams in the paper are very well done and illustrate the described methods appropriately. Figure 1 in particular was extremely helpful in understanding the approach.

The problem of text-to-graph generation based on functional requirements is a problem area that is underexplored, and this paper is a good contribution to it. In particular, the reformulation of PCQM4M is a helpful new resource for this task that this paper provides.

**Weaknesses:**

- The paper does not explain the reasoning for formulating the training objective as it is in Eqn 1. This is surprising given that experiments in Appendix D show a difference when using the differing term, but no satisfactory reason was given for why this might be the case.

- The experimental results are not very convincing of the importance of the message passing layer, as the SGG-LLM experiments with message passing enabled are not statistically significantly better than with message passing disabled (with reference to Table 2).

**Questions:**

**Questions**:

1. Baseline without fine-tuning: Under Appendix D, how was the SSG-LLM w/out fine-tuning prompted? Was there any investigation done using in-context learning / few-shot prompting to get a parsable generation?

2. As per Figure 1, is the reason that graph masking includes a self node (node i attends to node i) the same reason that causal masking has a token attending to itself? Is there any specific reason not to exclude that self node?

3. What is the reasoning for the formulation of the denominator in Eqn 1, and why must it differ from Eqn 2 for training on serialized graph sequences? Seems there is a missing explanation in the paragraph right under Eqn 2.

4. The feature vector for a node vector was selected as the feature vector of the last element describing that node. Were there any experiments done to vary this, for e.g. mean-pooling?

5. Why was BLOOM selected as the LLM backbone?

6. With reference to Table 3, why was parsability affected when using message passing?

7. With reference to Table 2, it seems that much of the improvement of SGG-LLM was in fine-tuning on the QED/Valency dataset. Moreover, the MAE on the QED dataset shows that the performance improvement when enabling message passing is not statistically significant (0.036 +/- 0.005, vs 0.044 +/- 0.011). Can the authors please highlight arguments for why the message passing inductive bias is working correctly, and what might be the issue in the current approach?


**Suggestions**:
- In Section 1, the paragraph on causal masking seems out of place. Suggest moving it to Section 3.1 before introducing causal graph masking.
- It is difficult to distinguish between the colours used in Tables 2 and 3. Would suggest finding a more accessibility-friendly solution, like using icons to distinguish between rows.

---

> ### Author Response · Authors · 2023-11-16
>
> [Reviewer] The paper does not explain the reasoning for formulating the training objective as it is in Eqn 1. This is surprising given that experiments in Appendix D show a difference when using the differing term, but no satisfactory reason was given for why this might be the case.
>
> [Authors] Thank you for pointing out the lack of clarity in our explanation. Upon reflection, it would have been best to introduce Eqn 2 first (the standard autoregressive training objective), then explain its deficiency with respect to the structure of the data and then introduce Eqn 1 (how we are proposing to train the model) as a objective which does taken into the structure of the data. This could make it clearer the reason for formulating Eqn 1. Instead we introduce Eqn 1 and then Eqn 2 and explain the difference between them with respect to the structure of our data.  Do you think this approach would improve the clarity of the objective?

---

> ### Author Response · Authors · 2023-11-16
>
> [Reviewer] Baseline without fine-tuning: Under Appendix D, how was the SSG-LLM w/out fine-tuning prompted? Was there any investigation done using in-context learning / few-shot prompting to get a parsable generation?
>
> [Authors] SGG-LLM w/out fine-tuning used the same prompts as with fine tuning, which requested a molecule with a specific functional property. Please note, models that are performant at in-context learning like GPT-4 or Llama-2 were too large to fine tune given cost and hardware constraints. However, qualitative experiments were performed to see if it was possible for GPT-4 to generate molecules with a requested number of valence electrons and it was only able to produce molecules which that consisted of a single atom. We will do our best to extend  the results to include performant in-context learning LLMs for the camera ready if given the opportunity.

---

> ### Author Response · Authors · 2023-11-16
>
> [Reviewer] As per Figure 1, is the reason that graph masking includes a self node (node i attends to node i) the same reason that causal masking has a token attending to itself? Is there any specific reason not to exclude that self node?
>
> [Authors] We think there is a conceptual reason which is not very strong and an experimental design reason which is strong. The conceptual reason is that if the self node is excluded, so is its representation from the new representation calculated by the attention layer. So some information about the node itself is excluded from the new representation. This argument is weakened by the fact that most message passing and attention layers contain skip connections, but still holds water. The experimental design reason is that all prior work includes the self node, so excluding it would be another element of the method we would have to justify. However the inclusion of a self node in the attention or message passing layers is not directly related to the problem this paper is addressing, so it would not strengthen the paper to include it.

---

> ### Author Response · Authors · 2023-11-16
>
> [Reviewer] The feature vector for a node vector was selected as the feature vector of the last element describing that node. Were there any experiments done to vary this, for e.g. mean-pooling?
>
> [Authors] No experiments were performed for other aggregation operations, but we chose this method because state-of-the-art sentence embedding techniques use the same approach. We leave it as future work to determine if there is a more performant aggregation method.

---

> ### Author Response · Authors · 2023-11-16
>
> [Reviewer] Why was BLOOM selected as the LLM backbone?
>
> [Authors] Prior work on serialized graph generation used the LLM T5, we found this model performed very poorly on the task of molecule generation. So we tried using BLOOM in experiments and it had good performance.

---

> ### Author Response · Authors · 2023-11-16
>
> [Reviewer] With reference to Table 3, why was parsability affected when using message passing?
>
> [Authors] Since sampling is a stochastic process and the difference in parsability is less than or equal 0.2% in all cases, it is hard to tell if message passing had any impact or if it was an artifact of the sampling process. All methods generated at least 998 /1000 parsable graphs. Even SGG-LLM w/o message passing did not have a parsability of 100% on one of the data sets, so a model without message passing also did not achieve 100% parsability.

---

### Meta-Review · Area_Chair_5NBq · 2023-12-09

**Metareview:**

The paper explores the novel problem of generating graphs conditioned on functional requirements through a text-to-text generation approach. The authors introduce a graph serialization/deserialization technique, a variant of negative log-likelihood objective, and interleaved message passing layers between transformer layers. They also conduct experiments on PCQM4M and WebNLG+ 2020 datasets to show the efficacy of the proposed approach.

Strengths: The paper is appreciated by reviewers for its contribution to an underexplored area—text-to-graph generation based on functional requirements. The diagrams, especially Figure 1, enhance the understanding of the proposed methods. The reformulation of PCQM4M is also recognized as a valuable resource for the task.

Several weaknesses are identified by the reviewers.
1. The paper lacks an explanation for the choice of the training objective in Equation 1, and the impact of this choice is not adequately addressed.
2. The experimental results regarding the importance of message passing layers are questioned, with concerns about statistical significance. Reviewer 4 also notes that the message passing idea is incremental.
3. Some reviewers point out the ill-defined task, inadequate exploration of baselines, and a limited technical contribution, particularly challenging the labeling of causal masks as a key contribution.

**Justification For Why Not Higher Score:**

While the paper makes a commendable contribution to an underexplored problem, it is crucial to address the identified weaknesses, provide clearer explanations for methodological choices, and conduct further analysis to validate the significance of proposed contributions. Improved clarity in presentation (pointed out by 3 out of 4 reviewers), addressing conflicting information, and refining language would enhance the overall quality of the paper.

**Justification For Why Not Lower Score:**

n/a

---

### Decision · Program_Chairs · 2024-01-16

Reject